Subject Areas:
behaviour

Keywords:
hawksbill turtle, *Eretmochelys imbricata*, leatherback turtle, *Dermochelys coriacea*, nesting behaviour

Authors for correspondence:
Thomas J. Burns
e-mail: tomb-09@hotmail.co.uk
Malcolm W. Kennedy
e-mail: malcolm.kennedy@glasgow.ac.uk

†Present address: Centre for Integrative Ecology, School of Life and Environmental Sciences, Deakin University (Burwood campus), Geelong, 3216, Australia.

# Buried treasure—marine turtles do not 'disguise' or 'camouflage' their nests but avoid them and create a decoy trail

Thomas J. Burns†, Rory R. Thomson, Rosemary A. McLaren, Jack Rawlinson, Euan McMillan, Hannah Davidson and Malcolm W. Kennedy

Institute of Biodiversity, Animal Health and Comparative Medicine, and School of Life Sciences, Graham Kerr Building, College of Medical, Veterinary and Life Sciences, University of Glasgow, Glasgow G12 8QQ, UK

TJB, 0000-0003-0408-8014; MWK, 0000-0002-0970-5264

After laying their eggs and refilling the egg chamber, sea turtles scatter sand extensively around the nest site. This is presumed to camouflage the nest, or optimize local conditions for egg development, but a consensus on its function is lacking. We quantified activity and mapped the movements of hawksbill (*Eretmochelys imbricata*) and leatherback (*Dermochelys coriacea*) turtles during sand-scattering. For leatherbacks, we also recorded activity at each sand-scattering position. For hawksbills, we recorded breathing rates during nesting as an indicator of metabolic investment and compared with published values for leatherbacks. Temporal and inferred metabolic investment in sand-scattering was substantial for both species. Neither species remained near the nest while sand-scattering, instead moving to several other positions to scatter sand, changing direction each time, progressively displacing themselves from the nest site. Movement patterns were highly diverse between individuals, but activity at each sand-scattering position changed little between completion of egg chamber refilling and return to the sea. Our findings are inconsistent with sand-scattering being to directly camouflage the nest, or primarily for modifying the nest-proximal environment. Instead, they are consistent with the construction of a series of dispersed decoy nests that may reduce the discovery of nests by predators.

# 1. Introduction

Sea turtles lay large clutches of eggs (50–130 depending on species [1]) in nest cavities in tropical or sub-tropical sand beaches, but provide no subsequent parental care. Their nesting processes comprise a series of discrete behavioural phases that are, at least superficially, consistent across species. There is variation in their description and categorization within and between species [1], but they broadly follow these steps: emergence from the sea and movement up the beach, selection of nesting site, preparation of nest site (body pitting), excavation of egg chamber, oviposition (egg laying), refilling of the egg chamber, scattering or throwing sand around the nest site (usually referred to as the 'camouflaging' or 'disguising' stage) and then return to the sea (e.g. [1–3]). We here deal with the inferred function of this sand-scattering.

Nest site selection and the excavation, and refilling of the egg chamber, are probably crucial to the reproductive success of sea turtles given the vagaries of weather and wave action [4], the need for suitable egg incubation conditions [5–7] and the escape of hatchlings to the surface [8]. Buried sea turtle egg clutches are also vulnerable to a range of predators [9,10] and dipteran parasite attack [11–13], with nest predation rates of more than 50% occurring on some beaches [14–16]. The mitigation of these risks is the most commonly presumed reason for the prolonged sand-scattering phase of nesting that follows completion of egg chamber refilling [3,17,18]. Other authors instead consider that its primary function is to establish beach conditions above and adjacent to the egg chamber for optimal incubation conditions [1,19]. Others still hint at a decoy function [20], but few studies have quantitatively examined sea turtle nesting behaviour (aside from nest site location and nesting frequency) in detail [21–23] and fewer still have focused on sand-scattering behaviour [24]. As such, these hypotheses remain essentially unexplored.

For leatherback turtles, *Dermochelys coriacea*, we previously reported that this last phase of nesting tends to be the longest, and potentially the most metabolically expensive [24]. Moreover, leatherback females move considerable distances away from their nests, stopping periodically to scatter sand while tracing a convoluted route that rarely if ever impinges upon the nest itself. We consequently used the neutral, non-presumptive term 'sand-scattering' for this phase because the turtles cannot be disguising or camouflaging a nest that they are not near.

Leatherbacks are distinctive among sea turtles—they are by far the largest, and have structural, physiological, anatomical, migratory, behavioural and dietary adaptations that distance them from other species [25], from which they separated about 100–150 Ma [26]. For example, they do not have rigid scutes, which could be among several adaptations permitting them to dive to depths of more than 1200 m [27], they spend prolonged periods in cold waters [28] and their gait on land as hatchlings and adults is distinct from other species [29]. This prompts the question of whether the sand-scattering behaviour pattern exhibited by leatherbacks is exclusive to them, or is avoidance of the nest during the final phase of nesting a common feature among sea turtles?

We now compare the sand-scattering phases of the smaller, hard-shelled, hawksbill turtle, *Eretmochelys imbricata*, with that of leatherbacks. Our observations suggest consistency in function across these two species and indicate that the prolonged sand-scattering phase after completion of activity at the nest is not to disguise or further camouflage their nests, or primarily to establish local beach conditions favourable to incubation of the eggs. Instead our findings are consistent with the idea that sea turtles establish a series of decoy nests that may distract and confuse egg-seeking predators and parasites, and increase the effort required to find and exploit a nest.

# 2. Material and methods

## 2.1. Study sites

Observations of nesting *E. imbricata* were made from June to August 2014, 2015, 2017 and 2018 on Campbellton and Hermitage Bays in the north east of Tobago (11.3172° N, 60.5600° W and 11.3162° N, 60.5708° W, respectively), between 19.30 h and 03.00 h. We observed *D. coriacea* between 19.00 h and 01.00 h on Fishing Pond beach on the east coast of Trinidad (10.5932° N, 61.0240° W) during June and July 2014 and 2019.

## 2.2. Hawksbill turtle nesting phase duration and breathing rates

We discriminated nesting phases following the descriptors given by Johnson *et al*. [17] and Burns *et al*. [24]. Hawksbills are prone to abandoning nesting if they detect disturbance nearby, so we recorded

only the excavation, oviposition, refilling and sand-scattering phases (see electronic supplementary material, appendix S1 for full description). We used an infrared illuminating night viewer (Yukon NV 5 × 60 Night Vision, Yukon Advanced Optics Worldwide, Vilnius, Lithuania) to observe the nesting phases and breathing rates. Intervals between breaths were taken from when a turtle's head reached its highest point in a breath until the next highest point.

## 2.3. Hawksbill and leatherback turtle sand-scattering movements

We placed markers (thin sticks with light-reflective tape) into the sand directly above the egg chamber when refilling was complete, and then at the position of the posterior tip (leatherbacks) or notch (hawksbills) of the carapace when a turtle began moving away from a sand-scattering position (here named 'stations'). Turtles rarely crossed over their own paths and our markers protruded above the level of sand thrown, such that the positions were rarely disturbed and never covered. If a marker was disturbed, we replaced it in the same position once the disturbance had passed. During sand-scattering, we made field sketches noting the relative position of markers and the path of animals between them. When the turtles had returned to sea, we measured the distances and angles of turn between the marked stations (the latter using a smartphone digital compass) and subsequently produced a scale drawing of paths travelled. For examples of hawksbill and leatherback scale drawings see electronic supplementary material and [24].

To assess whether there is a change in effort in sand-scattering as the phase progresses, we recorded time spent and the number of flipper actions (front and rear) at each station for leatherback turtles surveyed in 2019. For a front flipper action, we counted a bilateral sweep from their resting position to the side of the turtle forward then back as a single action. For rear flippers, we took the starting position as when the flippers are directly behind the turtle in resting position, and counted the movement of paired flippers to a side and then back to the centre point as a one movement. When both flippers then swept to the opposite side and returned to the centre point, we counted it as a second movement. We numbered sand-scattering stations sequentially, with sand-scattering carried out above the nest site designated station one.

## 2.4. Statistical procedures

We used R software (https://www.r-project.org) for all statistical analyses and graphing. We used linear mixed effects models (lme; nlme package [30]) to analyse phase duration and respiration rate during nesting data. All possible combinations of global model were compared (dredge; MuMIn package [31]) and then ranked using second-order Akaike information criterion ($AIC_c$). For our phase duration dataset, our global model included behavioural phase, nesting beach and year as categorical fixed effects, curved carapace width and curved carapace length as continuous fixed effects, and individual as a random effect. Two-way interactions between phase and each of the other fixed parameters were also included and a variance structure (varIdent; nlme package) was applied to behavioural phase to account for heterogeneity in residuals. We centred curved carapace width and length data (scale; base package [32]) prior to modelling. Our respiration dataset global model was identical, with the exception that curved carapace length was not included due to high correlation with curved carapace width ($r = 0.66$, $p < 0.001$). This correlation was only moderate ($r = 0.49$, $p < 0.001$) in our phase duration dataset, so both were included as parameters. We examined residual plots to check that assumptions about the distribution of errors were reasonable.

All fixed effects, interactions and individual, as a random effect, were included in the only highly ranked model ($\Delta AIC_c < 2$) for hawksbill nesting phase duration (electronic supplementary material, table S1). Behavioural phase of nesting and individual, as a random effect, were included in the highly ranked model of hawksbill respiration rate (electronic supplementary material, table S1). We made pairwise comparisons, with a Tukey adjustment, across the levels of fixed effects and interactions (emmeans and emtrends; emmeans [33]) included in each highly ranked model, and examined associated parameter estimate confidence intervals to determine the influence of fixed effects.

A Spearman rank correlation test (cor.test; stats package [32]) was used to examine the relationship between cumulative distance travelled and distance from the nest site at the end of the sand-scattering phase. To account for ties in the data, we confirmed $p$-values using 10 000 Monte Carlo re-sampling simulations (spearman_test; coin package [34]). To examine potential population turning preference during the sand-scattering phase, we first determined the proportion of turns to the right (right turns/ total turns) and the proportion of the total rotation to the right (degrees rotated right/total degrees rotated). We used a two-tailed one-sample $t$-test (t.test, stats package) to assess departure from chance.

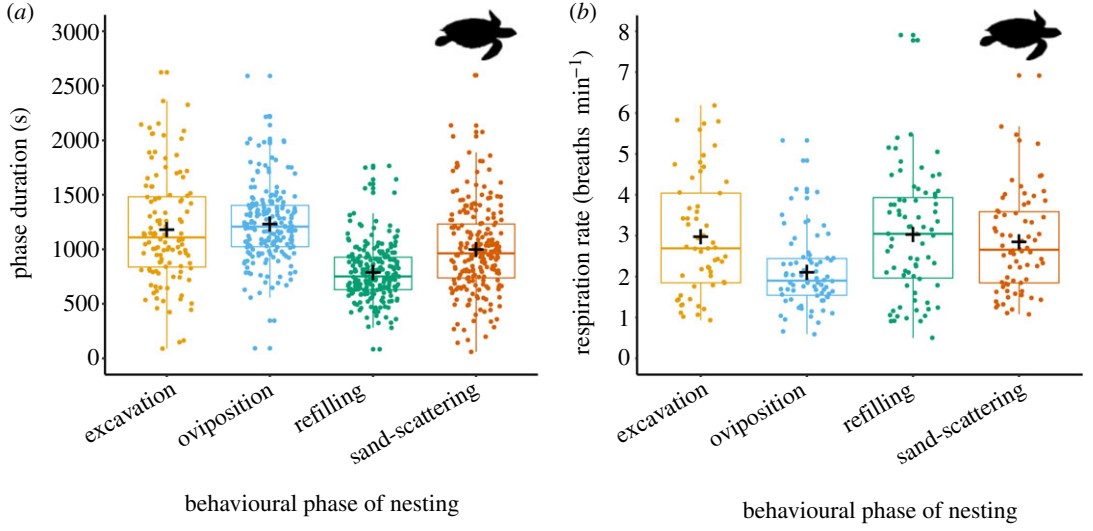

**Figure 1.** Duration of nesting phases, and breathing rates during them, for hawksbill turtles (*Eretmochelys imbricata*). (*a*) Data points indicate durations of complete nesting phases observed for individual turtles. (*b*) Changes in breathing frequency during the nesting process. Each point represents data from individual turtles averaged for breathing rates within each phase. Data for both panels from 2017 and 2018 seasons. Plots show the mean (+), median (horizontal line in the boxes), interquartile ranges (box boundaries; 75, upper and 25, lower percentiles), and the largest and smallest values within 1.5 × interquartile range below and above the 25th and 75th percentiles, respectively (whisker bars). Turtle silhouettes here and elsewhere obtained from http://phylopic.org/.

Hawksbill proportion of turn and total rotation to the right data was Box Cox transformed ($\lambda = 0.9$) prior to use.

We used a negative binomial mixed effects model (glmer.nb; lme4 [35]) to assess relative effort (total flipper actions) of leatherback turtles across sand-scattering stations. We included sand-scattering station and the total number of stations at which each turtle scattered sand as continuous explanatory variables, a two-way interaction between the former, and individual as a random effect. Total front and rear flipper actions at each station were recorded, but were found to be highly correlated ($r = 0.73$, $p < 0.001$), so the combined total was used in analyses. Time spent and total flipper actions at each scattering station were also highly correlated ($r = 0.85$, $p < 0.001$), so we used only total flipper actions as a response variable. We chose the latter as an assumed better proxy for metabolic effort and turtle impact on beach topography at a station. Station and total number of station values were re-scaled (scale function; R base package [32]) prior to use. We examined simulated residual plots (DHARMa package [36]) to check that assumptions about the distribution of errors were reasonable.

# 3. Results

## 3.1. Investment—nest phase durations and breathing rates—hawksbills

We previously examined the nesting phase durations and breathing rates of leatherback turtles during all phases of nesting [24], and here report on the same for hawksbill turtles. For hawksbills, refilling the nest hole was the shortest phase of nesting recorded, oviposition was the longest, and the sand-scattering phase was intermediate. The excavation phase was most variable and similar in duration to both oviposition and sand-scattering phases. The 95% confidence intervals for pairwise comparisons between refilling and all other phases (maximum range, −115 to −560.1), and between oviposition and sand-scattering (128.1–366) did not cross zero, suggesting strong differences between these phases (figure 1*a*; electronic supplementary material, table S2). For variations in phase durations between and within each year(s), and the slight influence of curved carapace width on phase duration see electronic supplementary material, tables S3 and S4, and figure S2. Curved carapace length and nesting beach were not strong predictors of the duration of any phase (95% confidence intervals cross zero, electronic supplementary material, tables S4 and S5).

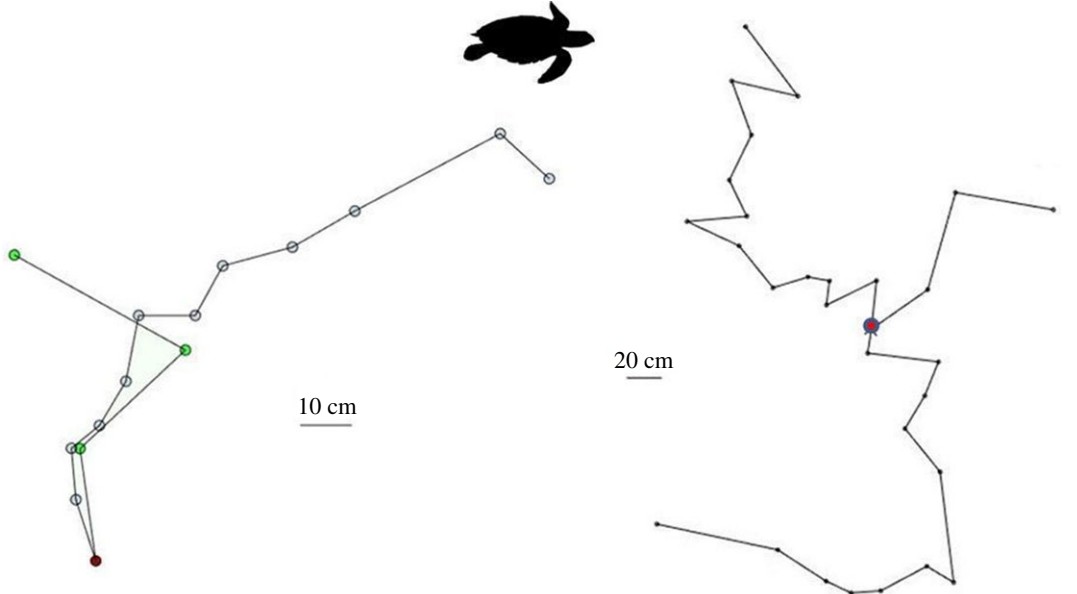

**Figure 2.** Sand-scattering track maps of hawksbill turtles after refilling the egg chamber and before departure to the sea. Examples of the tracks formed when separate individual hawksbill turtles return twice or three times in a season. The nest positions were at different locations on the beach but are here coincided to illustrate how differently they moved from their nests before returning to the sea. The superimposed position of the nest is indicated by a red-filled circle, the sand-scattering stations by circles. We have provided further example maps for individual hawksbill turtles recorded during the study in electronic supplementary material, figures S1 and S3 for leatherback turtles. Turtle silhouette not to scale.

Respiration rates were lowest during oviposition, but similar across excavation, refilling and sand-scattering. The 95% confidence intervals for pairwise comparisons between oviposition and all other phases did not cross zero (maximum ranges, 0.32 to 1.3 and −1.01 to −0.41), suggesting a strong effect. Confidence intervals for all other phase pairwise comparisons crossed zero (figure 1b; electronic supplementary material, table S6).

The sand-scattering phase of nesting by hawksbill turtles is, therefore, similar in duration and inferred metabolic effort to the other phases of nesting that involve movement and physical effort, namely excavation and refilling of the egg chamber, as is also the case for leatherback turtles [24].

## 3.2. Security—sand-scattering paths of hawksbill and leatherback turtles

After refilling the egg chamber, both species first scatter sand at the nest site, then move away from it, stopping at 'stations' to repeat sand-scattering actions before moving on, usually in a new direction, to repeat this process (figure 2 for movement maps for two turtles that returned repeatedly, and electronic supplementary material, figures S1 and S2 for 10 example maps recorded for each species). While movements at each station were similar, sand-scattering behaviour was otherwise highly variable across individuals, in both species. The number of scattering stations varied from 1 to 14 (mean = $6.95 \pm 3.22$ s.d., mode = 5, $n = 60$) for hawksbills (figure 3a); and from 5 to 24 (mean = $10.98 \pm 3.9$; mode = 10, $n = 54$) for leatherbacks (figure 3b). The distances between the scattering stations also ranged dramatically, from 6 to 147 cm (mean = $31.79 \pm 17.39$ cm, $n = 61$ individuals) for hawksbills; and from 22 to 318 cm (mean = $102.81 \pm 50.5$ cm, $n = 55$ individuals) for leatherbacks (figure 3c and d).

Leatherback turtles moved a greater mean distance between stations while sand-scattering in 2014 than 2019 (means = 145 and 80 cm, respectively; $t$-test, $t = 11.28$, $p < 0.001$), but stopped at a similar number of sand-scattering stations (Mann–Whitney $U$-test, $p = 0.76$, $W = 329.5$). We have found no methodological change to explain this and the difference could lay in reduced nest site space caused by the considerable beach erosion occurring on this coast (M.W. Kennedy 2014–2019, personal observations; [37]). Mean distance travelled between stations by hawksbills also varied between years, though by a much smaller margin (2017 and 2018 means = 30 and 40 cm, respectively; $t = 2.69$, $p = 0.01$). Hawksbill turtles also scattered sand at fewer stations in 2017 than 2018 (means = 5.6 and 8.6, respectively; $t = -3.67$, $p < 0.001$).

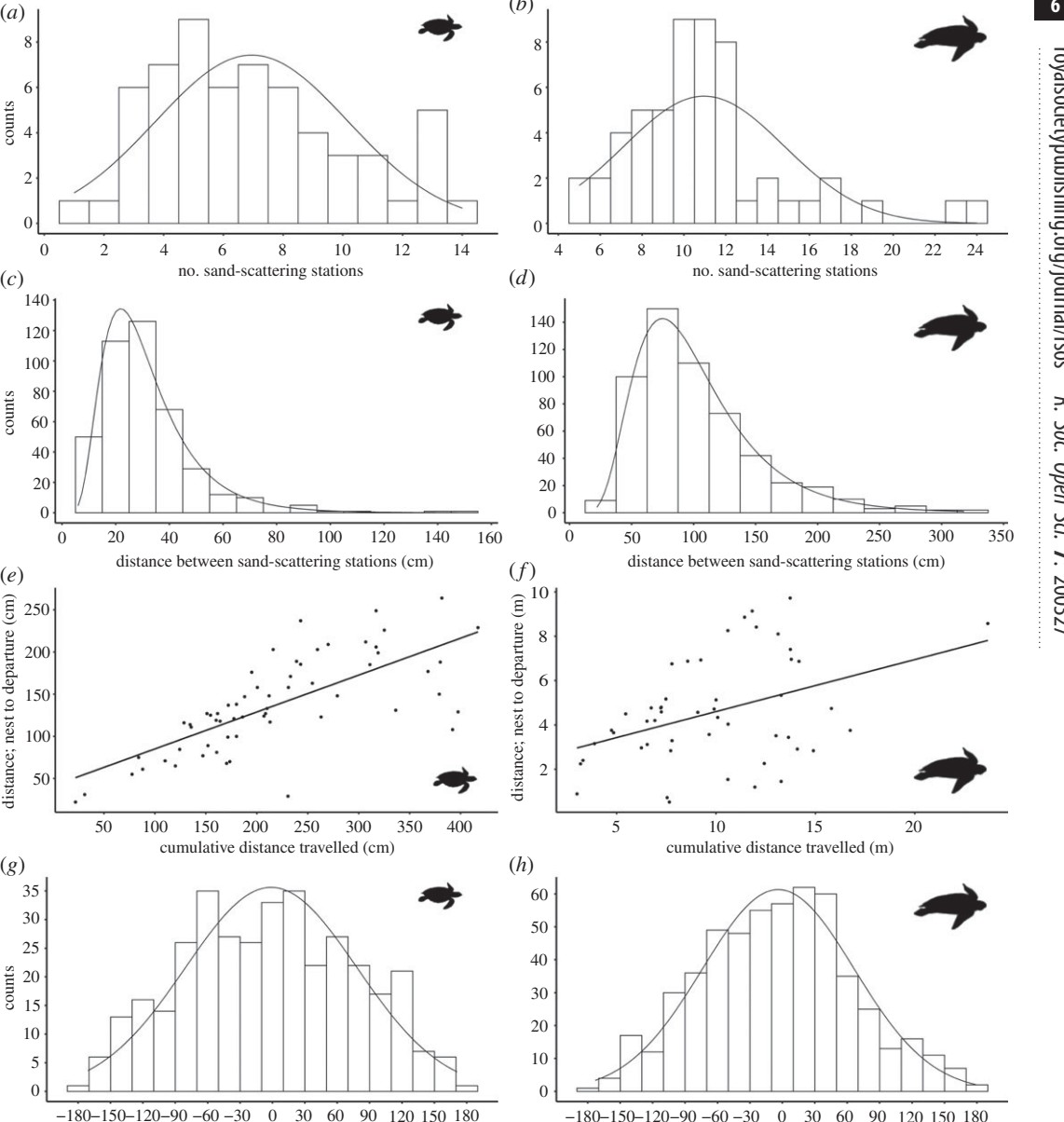

**Figure 3.** Parameters of the sand-scattering phases of hawksbill and leatherback sea turtles. Hawksbills—(a), (c), (e) and (g); leatherbacks—(b), (d), (f) and (h). (a) and (b), number of sand-scattering stations with normal/Gaussian curve fits; (c) and (d), distances between stations with lognormal curve fits; (e) and (f), the relationship between the cumulative distance between stations and final displacement distance between nest site and point of departure to the sea for each turtle, linear regression line fitted as guidance; and (g) and (h), overall population distributions of angles turned from one station to another.

For hawksbills, there was a strong correlation between summed distances between scattering stations and distance between the nest and the final departure point ($r_s = 0.77$, $n = 61$, $p < 0.0001$; figure 3e), indicative of movement progressively away from the nest. For leatherbacks, this correlation was substantially weaker ($r_s = 0.33$, $n = 50$, $p = 0.02$; figure 3f).

In both species the angles of turn between scattering stations varied from no change in bearing to an almost complete reversal. There was no population level preference in the direction of turn (hawksbills, $t_{59} = -0.44$, $p = 0.67$ or leatherbacks, $t_{53} = -0.58$, $p = 0.566$; figure 3g and h). However, leatherbacks showed a population level preference for turning a greater extent to the left that was not evident for hawksbill turtles ($t_{53} = -2.86$, $p = 0.006$ and $t_{59} = -0.76$, $p = 0.45$, respectively; figure 3g and h). Changes in direction between sand-scattering stations ranged remarkably among individuals (electronic supplementary

material, figures S1 and S3), and this also applied for two individual hawksbills that we observed nesting twice or three times in the same season (figure 2).

## 3.3. Consistency and diversity of activity at sand-scattering stations and movements between leatherbacks

Leatherback turtle activity (flipper actions) at sand-scattering stations did not change overall as the sand-scattering phase progressed (station number; $z_{318} = -0.111$, $p = 0.912$, $n = 30$ individuals). However, leatherback turtles that scattered sand at a greater number of sand-scattering stations were found to be less active at each station and consistently so (total number of stations; $z_{318} = -6.034$, $p < 0.0001$; figure 4). We found no interaction between sand-scattering station and the total number of sand-scattering stations ($z_{318} = 0.737$, $p = 0.461$).

# 4. Discussion

Hawksbill and leatherback turtles invest substantial temporal and inferred metabolic effort into sand-scattering behaviour after refilling the egg chamber (figure 1; [24]). We found that the paths that both species travel during this phase were not localized to the vicinity of the nest site, but instead became progressively displaced from it. While movements during the previous nesting phases are consistent and predictable, movements of both species during the sand-scattering phase were not. Both species engage in sand-scattering flipper actions, initially above the nest cavity immediately after refilling, and then at stations increasingly offset from it. However, here the consistency between individuals of a species ends; the number of scattering stations, distances between them, displacement between nest and point of departure, and the angles of turn at each scattering station were highly diverse (figure 3). Furthermore, no two turtles showed similar movement patterns, which was even evident between separate nestings by individuals observed on more than one occasion (figure 2, and electronic supplementary material, figures S1 and S3). The earlier, highly behaviourally consistent phases (excavation, oviposition, refilling of the egg chamber) probably function to optimize incubation conditions for eggs, protection from the elements, and emergence of hatchlings through overlaying layers of sand [1,4–8]. However, what might be the function of a prolonged, highly variable, seemingly disorganized, final part of nesting that involves considerable time, metabolic investment and extended exposure to predation and other risks?

Two main hypotheses on the primary function of sand-scattering behaviour have been suggested within the peer-reviewed literature. First, to camouflage or disguise the nest site position, and, second, to (re-)establish the beach environment around the nest to optimize temperature and moisture conditions for egg development [1,19]. Additionally, and more rarely, sand-scattering has been described as 'decoy' behaviour [20], in particular within the popular science literature [38]. However, despite the disparate views on its function, we are not aware of any studies that have explored these hypotheses in detail or quantitatively (excepting our own [24]). The camouflage or disguise hypothesis has intuitive appeal, but, given that the outcome of station-to-station sand-scattering is a greater area of disturbance [24], then this would make the vicinity of a nest site more, not less, obvious. Furthermore, some predators are indeed able to identify the position of nesting areas [39–41]. With regard to the second hypothesis, our observations of prolonged sand-scattering at positions progressively displaced from the nest site are not consistent with a primary function of creating optimal beach sand conditions for incubation proximal to a nest. Although the scattering activity immediately over the nest following completion of refilling the egg chamber might serve this function.

The sand-scattering behaviour we quantified is likely to disperse and de-localize indicators of a nest's position and recent presence of a nesting turtle, such as disturbed sand topography, texture and olfactory cues [39,40]. Furthermore, the very heterogeneity and unpredictability of this behaviour may operate to prevent predators from discerning a pattern of disturbance in the sand and olfactory cues to locate a nest [9,39]. For instance, were a female to scatter sand only in the immediate area around the nest, then a predator may detect a focus by sensing a gradient in such cues towards a central nest location.

It, therefore, seems more likely that the primary function of this behaviour, in leatherback and hawksbill turtles at least, is to leave behind a series of nest-like decoy disturbances in the sand. These decoys, while making the general nesting area more obvious, may confuse predators as to the exact position of a nest and increase the search and excavation effort required to find it. Increased search and excavation costs have been shown to alter nest predator foraging behaviour and predation risk [39,42,43]. This behaviour would be most effective against nest raiders that primarily detect nests from the tactile, visual and

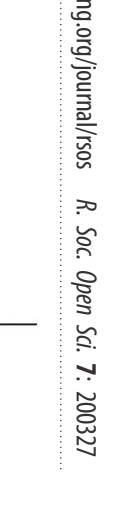

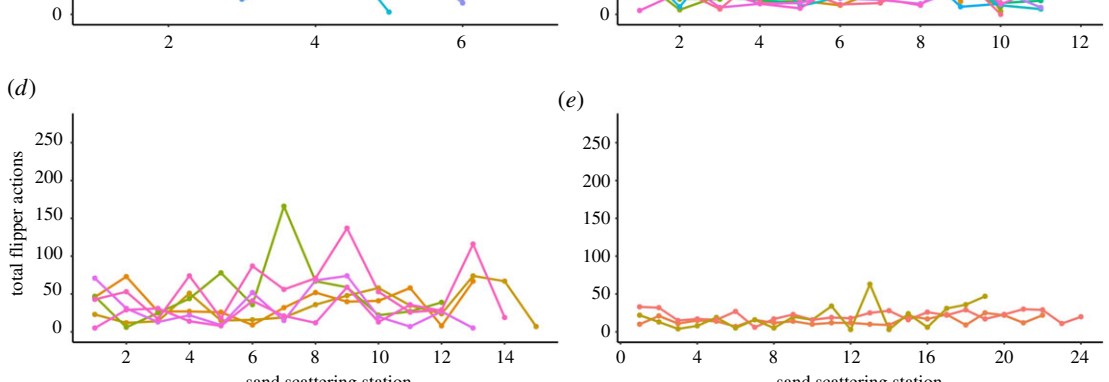

**Figure 4.** Flipper action activities of leatherback turtles during the sand-scattering phase of nesting that immediately follows the refilling of the egg chamber. Each line represents an individual turtle, with each point representing total flipper actions at a sand-scattering station (numbered sequentially as phase progressed, with '1' being immediately above the egg chamber). Panel (*a*) shows all sampled turtles, while smaller panels show turtles grouped by the total number of stations at which they scattered sand as follows: (*b*), ≤7; (*c*), 8–12; (*d*), 12–15; and (*e*), greater than 15 stations. Total flipper actions are the summed total of front and rear flipper actions at a station (see Material and methods for full description).

chemosensory cues present around nests soon after laying [39]. For example, it may function better against predation by Asian mongooses (*Herpestes javanicus*), which predate most hawksbill nests soon after laying [9], than ghost crabs, which predate most loggerhead turtle (*Caretta caretta*) nests when eggs are hatching [44].

All the leatherback and hawksbill turtles that we observed carried out sand-scattering offset from the nest site, with heterogeneity in actions a consistent feature for both species. However, there are also some notable differences between them. Temporal investment in sand-scattering behaviour, relative to excavation and oviposition, was greater by leatherbacks [24] than hawksbills (figure 1*a*). Furthermore, leatherback paths while sand-scattering were more circuitous than taken by hawksbills, and there tended to be greater distances between their sand-scattering stations (figure 3*e* and *f*; electronic supplementary material, figures S1 and S3). The preference of hawksbills for nesting and sand-scattering close to or

within vegetation [8,45,46], as opposed to the open beach preferred by leatherbacks [47], may explain some of this variation, as might differences in body size relative to available beach area.

If the actions at sand-scattering stations are to create a series of effective decoy nests, we would predict that the quality of such decoys should not change between the nest and return to the sea or else a predator might detect a gradient towards the nest. It might also be expected that activity at scattering stations might decay as the females tire, but we found no evidence of this for leatherback turtles. There was heterogeneity in the time spent and the number of flipper movements at each station, but there were no signs of overall decay in effort by individuals (figure 4). Also, if the number of fore and hind flipper movements at each station are an indicator of the quality of work, then we found no significant change as this final phase of nesting progressed either. Not only is this consistent with the decoy trail hypothesis, but also indicates the importance of investing in a prolonged final phase of nesting to the turtles' reproductive success.

# 5. Conclusion

Our findings question conventional wisdom about the final phase of nesting by sea turtles, and what it is for. Not every behaviour has a selective advantage, so whether one is attributable to the behaviour we describe remains conjectural. There is no clear route by which to experimentally investigate a reproductive advantage to this presumed decoy behaviour because it would involve, for example, comparing egg predation rates between nests with sand-scattering and those in which it is artificially prevented, using species for which there is increasing conservation concern (as for all marine turtles) [48]. However, it may be feasible to compare predation rates between nests for which there were different extents or movement patterns during sand-scattering.

Before returning to the sea, two phylogenetically and ecologically distant species of sea turtle engage in a prolonged, complex, highly diversified sand-scattering behaviour at positions away from the nest. This represents a considerable investment in time and effort while prolonging exposure of the females to risk. Our findings indicate that this behaviour cannot be described simply as disguising or camouflaging their nests and it is likewise unlikely that its primary function is the creation of beach conditions optimal for development of the eggs proximal to the nest site. They engage in what could better be described as decoy or distraction behaviours that may apply to all species of sea turtle.

Ethics. All procedures and permissions to be on protected beaches and work on leatherback and hawksbill turtles were approved by the Wildlife Section of the Forestry Division of the Government of Trinidad and Tobago.

Data accessibility. All the data used for graphing and statistical analyses are provided in electronic supplementary information excel files entitled 'Hawksbill nesting data decoy Burns *et al*.' and 'Leatherback nesting data decoy Burns *et al*.'.

Authors' contributions. Study design: M.W.K. and T.J.B. Fieldwork: T.J.B., R.R.T., R.A.M., J.R., E.M., H.D. and M.W.K. Data analysis and processing: T.J.B. and M.W.K. T.J.B and M.W.K. wrote the paper and all authors approved the final version.

Competing interests. The authors declare that they have no competing interests.

Funding. The Carnegie Trust for the Universities of Scotland, The Percy Sladen Trust.

Acknowledgements. We thank the Wildlife Section of the Trinidad and Tobago Government for access to nesting beaches, and the beach patrolling teams in Fishing Pond in Trinidad, and North East Sea Turtles, Tobago, for their field support and assistance. We are particularly grateful to Sookraj Persad in Trinidad, and Devon Eastman and Andel Mackenzie in Tobago, for helping us in the field and for passing on their considerable experience on turtle nesting. We also thank members of the separate Glasgow University expeditions to Trinidad and Tobago from 2014 to 2019 for assistance in the field.

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
