## [Reviewer comments · Royal Society Open Science]

Review History

RSOS-190675.R0 (Original submission)

Review form: Reviewer 1

Is the manuscript scientifically sound in its present form?

Yes

Are the interpretations and conclusions justified by the results?

No

Is the language acceptable?

Yes

Is it clear how to access all supporting data?

Yes

Do you have any ethical concerns with this paper?

No

Have you any concerns about statistical analyses in this paper?

No

Recommendation?

Major revision is needed (please make suggestions in comments)

Comments to the Author(s)

General comments:

This manuscript by Burns et al. provides an interesting take on the post-oviposition behaviour of hawksbill (and to a lesser extent, leatherback) turtles. The data are presented in an easy to understand way, and the explanations are clear. I would suggest the incorporation of some of the supplementary material into the main body of the text (e.g. examples of the movement maps for both species; and include the full statistical analysis methodology). I also have two main concerns with this manuscript, the first I outline mainly in my comments in the discussion, that the authors mention sand scattering as a method for altering the temperature of the underlying nest, but then make no further discussion. Secondly, I'm not sure how well you can measure the angle of a turn to a new stations given the level of sand scattering around the site. This is particularly relevant for hawksbills, where they only move short distances between stations and presumably scatter sand over their own tracks.

Finally, the line numbers often didn't line up with the text and reset on every page, I attempted to make it clear to what I was referring to but please have continuing numbers in the future for easier reference.

Specific comments:

Abstract

Pg 2 Line 2: You introduce that there are phases but don't go into more detail, I think either outline these or remove that statement.

Pg 2 Line 4: Change 'nest hole' to 'egg chamber.'

Introduction

The introduction is relatively short. There are a few areas that could be expanded slightly to allow the reader to have a better understanding of the process and system. In particular, this is written for someone who has knowledge of the nesting and reproductive biology of sea turtles, an outline of the life history and a brief description of the nesting procedure; i.e. body pitting, egg chambering and nest filling would be beneficial.

Pg 2 Line 31: Define 'large' - i.e. 50-150 depending on species (with references)

Pg 2 Line 31: Rephrase the end of this sentence to outline that there is not parental care, rather than they 'desert' them.

Pg 2 Line 33: I think we can agree that this is probable rather than likely.

Pg 2 Line 39: Why are eggs clutches so vulnerable to these? Because of the lack of parental investment?

Pg 3 Line 2: Specify what some of these specializations are.

Methods

I am unsure as to why the full statistical methods are included as supplementary rather than included in the main body of the text. There's not substantially more included in the supplement and it should be included in the main text. Also - why was turtle width used instead of turtle length, and was this curved carapace or straight carapace measures?

Pg 3 Line 25: Either specify GPS coordinates or (in my opinion; preferentially) have a figure with the locations where the study took place.

Pg 3 Line 40: Again, here you talk about only observing certain parts of the nesting procedure but there's no previous mention of how this process occurs.

Pg 3 Line 50: Were these markers placed directly on top of the egg chamber? How were covering females kept from uprooting the marker?

Pg 3 Line 54: Given the level of sand-scattering that a turtle produces during the final phase of nesting, how did you follow paths that had been covered by the sand-scatter? Can you be sure that you could accurately follow the path of the turtle?

Pg 3 Line 56: Rather than have the reader go to two different sources for examples of scale drawings, place a leatherback version in the supplement as well.

Pg 4 Line 3: Make sure you cite the R development team: use the command 'citation()' in R to obtain the correct citation.

Pg 4 Line 6: You need to directly cite the paper associated with the R package rather than the CRAN mirror.

Pg 4 Line 14: Is there a reason that AIC selection wasn't employed for model fitting?

Results

Pg 4 Line 45: Do you distinguish between 'body pitting' and 'egg chambering' when referring to 'nest excavation'?

Pg 4 Line 60 – Pg 5 Line 4: This is a very broad statement, and belongs in the discussion rather than the results. I'm also not sure that you can make these claims without some sort of statistical comparison to support them. An analysis of the differences between species is therefore required.

Pg 5 Line 10: Again, suggest using the term 'egg chamber' rather than 'nest hole'.

Pg 5 Line 48: I do not like the constant referring back to reference 9 for the leatherback outputs. I think you either need to include some raw data and figures here to maintain the story, or simply present the hawksbill data and make the comparisons in the discussion.

Figure 2c & 2d: Make sure the X-axis labels are consistent.

Discussion

Overall, the discussion is quite brief and lacking citations in a number of areas. I think some more time needs to be spent on discussing the differences observed here with the observations made previously in the leatherbacks. I also believe that the nest decoy hypothesis has been raised previously, and it should be noted that this isn't a particularly novel hypothesis.

Pg 5 Line 60: Change physiological effort to metabolic effort – it's more about the energy expenditure than the underlying physiology.

Pg 6 Line 6: Change stereotypical to consistent

Pg 6 Lines 19 – 25: Need references here.

Pg 7 Line 1: You mention that the second hypothesis may be to produce favourable incubation conditions for the developing embryos, yet you make no mention of it in this paragraph.

Furthermore, your conclusion is that the purpose of sand scattering is to create decoys, but you have no evidence to disprove that it may be used for nest environment alterations.

Pg 7 Line 23: But you said there was inter-individual variation, could you not look at the effort in sand scattering and predation rates? You would expect to see lower predation rates in individuals that spend longer creating these 'decoys'.

Pg 7 Line 45: Again, I'm not sure you can make this conclusions without at least mentioning that the sand scattering could be related to nest environment manipulation...

Decision letter (RSOS-190675.R0)

13-Aug-2019

Dear Dr Kennedy:

Manuscript ID RSOS-190675 entitled "Buried treasure - Marine turtles do not 'disguise' or 'camouflage' their nests but avoid them and create a decoy trail" which you submitted to Royal Society Open Science, has been reviewed. The comments from reviewers are included at the bottom of this letter.

In view of the criticisms of the reviewers, the manuscript has been rejected in its current form. However, a new manuscript may be submitted which takes into consideration these comments.

Please note that resubmitting your manuscript does not guarantee eventual acceptance, and that your resubmission will be subject to peer review before a decision is made.

Your resubmitted manuscript should be submitted by 10-Feb-2020. If you are unable to submit by this date please contact the Editorial Office.

on behalf of Dr Ari Friedlaender (Associate Editor) and Kevin Padian (Subject Editor)
openscience@royalsociety.org

Associate Editor Comments to Author (Dr Ari Friedlaender):

To the Authors,
Thank you for submitting your work. I apologize for the delay in returning it. In general, this work is well done and of scientific interest. However, there are some fundamental features that are missing that require attention and that preclude accepting it in its current form. I and the reviewer agree that while the data are generally presented well, there are significant issues relating to citations throughout the manuscript that need to be added. As well, the introduction requires attention to place the current work in the context of previous/current knowledge in the field. Similarly, the discussion requires more rigor in how the results justify your conclusions and how the variables that were chosen may or may not affect your results. Given that there is general interest in the work, I would like to give you the opportunity to resubmit a revised version of your work in the future.

Please see the reviewer comments for more specific details and comments.

Thank you.
Ari S. Friedlaender

Reviewers' Comments to Author:
Reviewer: 1

Comments to the Author(s)
General comments:

This manuscript by Burns et al. provides an interesting take on the post-oviposition behaviour of hawksbill (and to a lesser extent, leatherback) turtles. The data are presented in an easy to understand way, and the explanations are clear. I would suggest the incorporation of some of the supplementary material into the main body of the text (e.g. examples of the movement maps for

both species; and include the full statistical analysis methodology). I also have two main concerns with this manuscript, the first I outline mainly in my comments in the discussion, that the authors mention sand scattering as a method for altering the temperature of the underlying nest, but then make no further discussion. Secondly, I'm not sure how well you can measure the angle of a turn to a new stations given the level of sand scattering around the site. This is particularly relevant for hawksbills, where they only move short distances between stations and presumably scatter sand over their own tracks.

Finally, the line numbers often didn't line up with the text and reset on every page, I attempted to make it clear to what I was referring to but please have continuing numbers in the future for easier reference.

Specific comments:

Abstract

Pg 2 Line 2: You introduce that there are phases but don't go into more detail, I think either outline these or remove that statement.

Pg 2 Line 4: Change 'nest hole' to 'egg chamber.'

Introduction

The introduction is relatively short. There are a few areas that could be expanded slightly to allow the reader to have a better understanding of the process and system. In particular, this is written for someone who has knowledge of the nesting and reproductive biology of sea turtles, an outline of the life history and a brief description of the nesting procedure; i.e. body pitting, egg chambering and nest filling would be beneficial.

Pg 2 Line 31: Define 'large' – i.e. 50-150 depending on species (with references)

Pg 2 Line 31: Rephrase the end of this sentence to outline that there is not parental care, rather than they 'desert' them.

Pg 2 Line 33: I think we can agree that this is probable rather than likely.

Pg 2 Line 39: Why are eggs clutches so vulnerable to these? Because of the lack of parental investment?

Pg 3 Line 2: Specify what some of these specializations are.

Methods

I am unsure as to why the full statistical methods are included as supplementary rather than included in the main body of the text. There's not substantially more included in the supplement and it should be included in the main text. Also – why was turtle width used instead of turtle length, and was this curved carapace or straight carapace measures?

Pg 3 Line 25: Either specify GPS coordinates or (in my opinion; preferentially) have a figure with the locations where the study took place.

Pg 3 Line 40: Again, here you talk about only observing certain parts of the nesting procedure but there's no previous mention of how this process occurs.

Pg 3 Line 50: Were these markers placed directly on top of the egg chamber? How were covering females kept from uprooting the marker?

Pg 3 Line 54: Given the level of sand-scattering that a turtle produces during the final phase of nesting, how did you follow paths that had been covered by the sand-scatter? Can you be sure that you could accurately follow the path of the turtle?

Pg 3 Line 56: Rather than have the reader go to two different sources for examples of scale drawings, place a leatherback version in the supplement as well.

Pg 4 Line 3: Make sure you cite the R development team: use the command 'citation()' in R to obtain the correct citation.

Pg 4 Line 6: You need to directly cite the paper associated with the R package rather than the CRAN mirror.

Pg 4 Line 14: Is there a reason that AIC selection wasn't employed for model fitting?

Results

Pg 4 Line 45: Do you distinguish between 'body pitting' and 'egg chambering' when referring to 'nest excavation'?

Pg 4 Line 60 – Pg 5 Line 4: This is a very broad statement, and belongs in the discussion rather than the results. I'm also not sure that you can make these claims without some sort of statistical comparison to support them. An analysis of the differences between species is therefore required.

Pg 5 Line 10: Again, suggest using the term 'egg chamber' rather than 'nest hole'.

Pg 5 Line 48: I do not like the constant referring back to reference 9 for the leatherback outputs. I think you either need to include some raw data and figures here to maintain the story, or simply present the hawksbill data and make the comparisons in the discussion.

Figure 2c & 2d: Make sure the X-axis labels are consistent.

Discussion

Overall, the discussion is quite brief and lacking citations in a number of areas. I think some more time needs to be spent on discussing the differences observed here with the observations made previously in the leatherbacks. I also believe that the nest decoy hypothesis has been raised previously, and it should be noted that this isn't a particularly novel hypothesis.

Pg 5 Line 60: Change physiological effort to metabolic effort – it's more about the energy expenditure than the underlying physiology.

Pg 6 Line 6: Change stereotypical to consistent

Pg 6 Lines 19 – 25: Need references here.

Pg 7 Line 1: You mention that the second hypothesis may be to produce favourable incubation conditions for the developing embryos, yet you make no mention of it in this paragraph.

Furthermore, your conclusion is that the purpose of sand scattering is to create decoys, but you have no evidence to disprove that it may be used for nest environment alterations.

Pg 7 Line 23: But you said there was inter-individual variation, could you not look at the effort in sand scattering and predation rates? You would expect to see lower predation rates in individuals that spend longer creating these 'decoys'.

Pg 7 Line 45: Again, I'm not sure you can make this conclusions without at least mentioning that the sand scattering could be related to nest environment manipulation...

Author's Response to Decision Letter for (RSOS-190675.R0)

See Appendix A.

RSOS-200327.R0

Review form: Reviewer 1

Is the manuscript scientifically sound in its present form?

Yes

Are the interpretations and conclusions justified by the results?

Yes

Is the language acceptable?

Yes

Do you have any ethical concerns with this paper?

No

Have you any concerns about statistical analyses in this paper?

No

Recommendation?

Accept with minor revision (please list in comments)

Comments to the Author(s)

General comments:

I thank the authors for taking the time to address my comments on their manuscript. I believe that the manuscript is clearer and more comprehensive following review and am happy that it is fit for publication following a few very minor changes (see below). For future reference, I encourage the authors to track their changes on the document or highlight regions that have been changed to facilitate a more effective way of gauging the changes that have been made in response to a reviewer's comments.

The inclusions of field sketches of the sand scattering stations are effective; however, they are excessive. I recommend keeping some 'type' examples that can be shown as examples and removing the rest.

Introduction

The introduction now reads much better, there are still a couple of missing references (detailed below), but I only have minor comments.

Lines 47-51: Insert a reference in regards to the nesting process, there should be one described in one of the volumes of *The Biology of Sea Turtles* if none others are available.

Methods

The methods are also clearer following the review, especially with the inclusion of the full statistical methodology in this section.

Lines 98-99: I appreciate the authors supplying the GPS coordinates for the study sites. I would recommend checking the GPS coordinates (especially 11.3166°N, -0.571031°E), keeping the number of decimal places consistent, and changing the longitudinal coordinates to 60.5600°W rather than having a negative Eastern coordinate.

Lines 146-148: I think if you are going to use one of CCL or CCW, CCL makes more biological sense to include in the model. However, given that the analyses are completed and re-analyzing would cause a delay, I am content with CCW being used, but it's something to consider in the future.

Results

The results are clear and concise and I have no further comments for improvement.

Discussion

Lines 275-278: I typically avoid and discourage the use of rhetorical questions, but leave it up to the discretion of the authors/editors as to whether it is rephrased into a statement.

Lines 289: Don't start a sentence with "And."

Decision letter (RSOS-200327.R0)

31-Mar-2020

Dear Dr Kennedy

On behalf of the Editor, I am pleased to inform you that your Manuscript RSOS-200327 entitled "Buried treasure - Marine turtles do not 'disguise' or 'camouflage' their nests but avoid them and create a decoy trail" has been accepted for publication in *Royal Society Open Science* subject to minor revision in accordance with the referee suggestions. Please find the referees' comments at the end of this email.

The reviewers and Subject Editor have recommended publication, but also suggest some minor revisions to your manuscript. Therefore, I invite you to respond to the comments and revise your manuscript.

- Ethics statement

- Data accessibility

<http://datadryad.org/submit?journalID=RSOS&manu=RSOS-200327>

- Competing interests

- Authors' contributions

- Acknowledgements

- Funding statement

Please note that we cannot publish your manuscript without these end statements included. We have included a screenshot example of the end statements for reference. If you feel that a given

heading is not relevant to your paper, please nevertheless include the heading and explicitly state that it is not relevant to your work.

Because the schedule for publication is very tight, it is a condition of publication that you submit the revised version of your manuscript before 09-Apr-2020. Please note that the revision deadline will expire at 00.00am on this date. If you do not think you will be able to meet this date please let me know immediately.

on behalf of Dr Ari Friedlaender (Associate Editor) and Kevin Padian (Subject Editor)
openscience@royalsociety.org

Reviewer comments to Author:

Reviewer: 1

Comments to the Author(s)

General comments:

I thank the authors for taking the time to address my comments on their manuscript. I believe that the manuscript is clearer and more comprehensive following review and am happy that it is fit for publication following a few very minor changes (see below). For future reference, I encourage the authors to track their changes on the document or highlight regions that have been changed to facilitate a more effective way of gauging the changes that have been made in response to a reviewer's comments.

The inclusions of field sketches of the sand scattering stations are effective; however, they are excessive. I recommend keeping some 'type' examples that can be shown as examples and removing the rest.

Introduction

The introduction now reads much better, there are still a couple of missing references (detailed below), but I only have minor comments.

Lines 47-51: Insert a reference in regards to the nesting process, there should be one described in one of the volumes of *The Biology of Sea Turtles* if none others are available.

Methods

The methods are also clearer following the review, especially with the inclusion of the full statistical methodology in this section.

Lines 98-99: I appreciate the authors supplying the GPS coordinates for the study sites. I would recommend checking the GPS coordinates (especially 11.3166°N, -0.571031°E), keeping the number of decimal places consistent, and changing the longitudinal coordinates to 60.5600°W rather than having a negative Eastern coordinate.

Lines 146-148: I think if you are going to use one of CCL or CCW, CCL makes more biological sense to include in the model. However, given that the analyses are completed and re-analyzing would cause a delay, I am content with CCW being used, but it's something to consider in the future.

Results

The results are clear and concise and I have no further comments for improvement.

Discussion

Lines 275-278: I typically avoid and discourage the use of rhetorical questions, but leave it up to the discretion of the authors/editors as to whether it is rephrased into a statement.

Lines 289: Don't start a sentence with "And."

Author's Response to Decision Letter for (RSOS-200327.R0)

See Appendices B & C.

Decision letter (RSOS-200327.R1)

06-Apr-2020

Dear Dr Kennedy,

It is a pleasure to accept your manuscript entitled "Buried treasure - Marine turtles do not 'disguise' or 'camouflage' their nests but avoid them and create a decoy trail" in its current form for publication in Royal Society Open Science.

Kind regards,

Lianne Parkhouse

on behalf of Dr Ari Friedlaender (Associate Editor) and Kevin Padian (Subject Editor)
openscience@royalsociety.org

Appendix A

Original manuscript - Manuscript ID RSOS-190675

Title - **Buried treasure - Marine turtles do not ‘disguise’ or ‘camouflage’ their nests but avoid them and create a decoy trail**

Authors - Thomas J. Burns, Rory R. Thomson, Rosemary A. McLaren, Jack Rawlinson, Euan McMillan, Hannah Davidson and Malcolm W. Kennedy

Dear Dr Friedlander,

We believe that we have improved and updated the manuscript that should hopefully now be to the satisfaction of the reviewers, and we respond to all of their points in detail below. We have added a new author, Jack Rawlinson, who provided us with new data that contributed to us satisfying the reviewers’ comments and reinforced the points made in the original submission.

Best wishes,

Malcolm Kennedy

On behalf of all authors.

Below is the original decision letter with text in italics, and our responses in plain text.

Associate Editor Comments to Author (Dr Ari Friedlaender):

To the Authors,

Thank you for submitting your work. I apologize for the delay in returning it. In general, this work is well done and of scientific interest. However, there are some fundamental features that are missing that require attention and that preclude accepting it in its current form. I and the reviewer agree that while the data are generally presented well, there are significant issues relating to citations throughout the manuscript that need to be added. As well, the introduction requires attention to place the current work in the context of previous/current knowledge in the field. Similarly, the discussion requires more rigor in how the results justify your conclusions and how the variables that were chosen may or may not affect your results. Given that there is general interest in the work, I would like to give you the opportunity to resubmit a revised version of your work in the future.

Please see the reviewer comments for more specific details and comments.

Thank you.

Ari S. Friedlaender

Reviewers' Comments to Author:

Reviewer: 1

Comments to the Author(s)

General comments:

This manuscript by Burns et al. provides an interesting take on the post-oviposition behaviour of hawksbill (and to a lesser extent, leatherback) turtles. The data are presented in an easy to understand way, and the explanations are clear. I would suggest the incorporation of some of the supplementary material into the main body of the text (e.g. examples of the movement maps for both species; and include the full statistical analysis methodology). I also have two main concerns with this manuscript, the first I outline mainly in my comments in the discussion, that the authors mention sand scattering as a method for altering the temperature of the underlying nest, but then make no further discussion. Secondly, I'm not sure how well you can measure the angle of a turn to a new stations given the level of sand scattering around the site. This is particularly relevant for hawksbills, where they only move short distances between stations and presumably scatter sand over their own tracks. Finally, the line numbers often didn't line up with the text and reset on every page, I attempted to make it clear to what I was referring to but please have continuing numbers in the future for easier reference.

We now include two diagrams of movement maps, specifically for the two hawksbill turtles that we observed on return visits, twice and thrice – please see figure 2 in the main text. The large number of other maps for hawksbills and the new ones for leatherbacks remain the Supplementary (figure S1 and S3).

A description of all statistical methods is now included in the main text.

We have added detail to the introduction and discussion on the nest environment hypothesis.

We are confident that we were able to measure turning angles with reasonable accuracy – please see below for more detail on what we did.

Specific comments:

Abstract

1) Pg 2 Line 2: You introduce that there are phases but don't go into more detail, I think either outline these or remove that statement.

We have removed this statement and added detail of nesting phases that should be sufficient as explanations in the main text. To avoid excess text and repetition from other papers, including our own, we have also included fuller descriptions in the Supplement Appendix 1.

2) Pg 2 Line 4: Change 'nest hole' to 'egg chamber.'

Changed as suggested.

Introduction

3) *The introduction is relatively short. There are a few areas that could be expanded slightly to allow the reader to have a better understanding of the process and system. In particular, this is written for someone who has knowledge of the nesting and reproductive biology of sea turtles, an outline of the life history and a brief description of the nesting procedure; i.e. body pitting, egg chambering and nest filling would be beneficial.*

Outline of the whole sea turtle nesting process has been added, as well as further details of the differences between the two species, and a clearer description of the nest environment hypothesis. This also emphasises the disparities between the two species in their ecologies, and, of more direct relevance to their activities on the beach, how leatherbacks' gaits on land are different, which could influence how they behave and move on their nesting beach relative to a distantly related species such as hawksbills.

4) *Pg 2 Line 31: Define 'large' – i.e. 50-150 depending on species (with references)*

Summary detail on egg numbers and citations have been added.

5) *Pg 2 Line 31: Rephrase the end of this sentence to outline that there is not parental care, rather than they 'desert' them.*

Re-phrased as suggested.

6) *Pg 2 Line 33: I think we can agree that this is probable rather than likely.*

Changed from 'likely to be' to 'probably'.

7) *Pg 2 Line 39: Why are eggs clutches so vulnerable to these? Because of the lack of parental investment?*

We have not been able to find any literature that discusses this directly in relation to turtles. Our understanding is that parental care is rare across Chelonii, probably particularly so for aquatic species. We can see it has relevance in relation to our findings on potential post-ovipositional parental care, but introducing this topic here may be distracting.

8) *Pg 3 Line 2: Specify what some of these specializations are.*

Additional text detailing diving, thermal tolerance, differences in gait on land, etc., has been added.

9) *Methods*

I am unsure as to why the full statistical methods are included as supplementary rather than included in the main body of the text. There's not substantially more included in the supplement and it should be included in the main text. Also – why was turtle width used instead of turtle length, and was this curved carapace or straight carapace measures?

We have added the full statistical methods to the main text.

Width and length were of the curved carapace. We have now clarified this in text. Turtle length (CCL) was included in the original global model for phase duration but erroneously omitted in the original submission – the text has been edited to reflect this. CCL was not included in our respiration rate global model, as it was highly correlated with CCW in this dataset.

10) Pg 3 Line 25: Either specify GPS coordinates or (in my opinion; preferentially) have a figure with the locations where the study took place.

We have added coordinates for each study site to the methods. Along with the descriptions of the sites, this should take the curious reader directly to the study beaches, so we have not included maps.

11) Pg 3 Line 40: Again, here you talk about only observing certain parts of the nesting procedure but there's no previous mention of how this process occurs.

Additional detail outlining the nesting phases has now been added to the Introduction.

12) Pg 3 Line 50: Were these markers placed directly on top of the egg chamber? How were covering females kept from uprooting the marker?

Additional detail has been added to this section, copied here as follows:

"We placed markers (thin sticks with light-reflective tape) into the sand directly above the egg chamber when refilling was complete, and then at the position of the posterior tip (leatherbacks) or notch (hawksbills) of the carapace when a turtle began moving away from subsequent sand-scattering position (here named 'stations'). Turtles rarely crossed over their own paths and our markers protruded above the level of sand thrown, such that the positions were rarely disturbed and never covered. If a marker was disturbed, we replaced it in the same position once the disturbance had passed."

13) Pg 3 Line 54: Given the level of sand-scattering that a turtle produces during the final phase of nesting, how did you follow paths that had been covered by the sand-scatter? Can you be sure that you could accurately follow the path of the turtle?

In addition to placing markers (see above), field sketches and notes of the relative position of sand scattering stations and the turtles' movements between were made by observers. This allowed us to accurately reproduce their path once they had returned to the sea.

14) Pg 3 Line 56: Rather than have the reader go to two different sources for examples of scale drawings, place a leatherback version in the supplement as well.

We have added further data (collected June to August 2019) on leatherback scattering actions to the manuscript. As part of this, ten scale drawings of leatherback scattering paths are included in the supplementary material (figure S3).

15) Pg 4 Line 3: Make sure you cite the R development team: use the command 'citation()' in R to obtain the correct citation.

and

16) Pg 4 Line 6: You need to directly cite the paper associated with the R package rather than the CRAN mirror.

Full citations for all R packages are now included.

Pg 4 Line 14: Is there a reason that AIC selection wasn't employed for model fitting?

We re-ran the analyses for phase duration and respiration rate using the same global models with the MuMIn: dredge function to compare all possible models and then ranked them by AICc. We then drew inference from highly ranked models ($\Delta AICc \leq 2$) by examining 95% confidence intervals for trends or pairwise comparisons computed using the emmeans package; emmeans and emtrends functions.

Results

17) Pg 4 Line 45: Do you distinguish between 'body pitting' and 'egg chambering' when referring to 'nest excavation'?

Yes. Additional detail on classification of nesting phases added to the Introduction and Methods to make this clear, plus fuller descriptions in Appendix S1 of the Supplement.

18) Pg 4 Line 60 – Pg 5 Line 4: This is a very broad statement, and belongs in the discussion rather than the results. I'm also not sure that you can make these claims without some sort of statistical comparison to support them. An analysis of the differences between species is therefore required.

We have removed this statement from the manuscript.

19) Pg 5 Line 10: Again, suggest using the term 'egg chamber' rather than 'nest hole'.

Changed as suggested.

20) Pg 5 Line 48: I do not like the constant referring back to reference 9 for the leatherback outputs. I think you either need to include some raw data and figures here to maintain the story, or simply present the hawksbill data and make the comparisons in the discussion.

Yes, this was an artefact of the dearth of publications dealing with quantitative analysis of sea turtle nesting phases. We have reduced reference to this paper and added further leatherback data to the main text so that points are not lost.

21) Figure 2c & 2d: Make sure the X-axis labels are consistent.

Axis labels are now consistent.

Discussion

22) Overall, the discussion is quite brief and lacking citations in a number of areas. I think some more time needs to be spent on discussing the differences observed here with the observations made previously in the leatherbacks. I also believe that the nest decoy hypothesis has been raised previously, and it should be noted that this isn't a particularly novel hypothesis.

We have added more context to the Discussion highlighting the differences between the species in presumed decoy behaviour, and also to deal with the extra new material of leatherbacks. With regard to the nest decoy hypothesis, yes, it has been mooted before, but only in a very limited way in the peer-reviewed literature as far as we can find. It does appear in some popular science articles, but never with any quantitative or investigative detail. Our Scopus and Google Scholar searches yielded very little that mention decoy behaviour other than our own previous publication. However, we have added text and citations on previous literature on the decoy idea.

23) Pg 5 Line 60: *Change physiological effort to metabolic effort – it's more about the energy expenditure than the underlying physiology.*

Changed as suggested.

24) Pg 6 Line 6: *Change stereotypical to consistent*

Changed as suggested.

25) Pg 6 Lines 19 – 25: *Need references here.*

References added.

26) Pg 7 Line 1: *You mention that the second hypothesis may be to produce favourable incubation conditions for the developing embryos, yet you make no mention of it in this paragraph. Furthermore, your conclusion is that the purpose of sand scattering is to create decoys, but you have no evidence to disprove that it may be used for nest environment alterations.*

We agree. Our findings do not disprove that sand scattering may play a role in creating optimal incubation conditions. The final sand scattering actions directly over the nest site following completion of refilling the nest cavity could indeed act to optimise the nest environment (which we now acknowledge in the text, and as a cited possibility raised in the literature), but this cannot easily be said of sand scattering at positions substantially and progressively away from the nest site. So, our deduction from what we observe is that the extensive area sometimes metres beyond the nest over which the females scatter sand is unlikely to modify the nest environment. The favourable incubation hypothesis is, however, still open to investigation and debate, perhaps now in a modified form. We trust, however, that our amendments to the Discussion adequately cover the hypothesis and how we interpret our observations with reference to it.

27) Pg 7 Line 23: *But you said there was inter-individual variation, could you not look at the effort in sand scattering and predation rates? You would expect to see lower predation rates in individuals that spend longer creating these 'decoys'.*

Yes, that would indeed be a prediction and we would very much like to investigate this. It would require a considerable upscaling of the work and would be complicated by nesting turtles working over and around the tracks of predecessors. It is possible, for instance, that those females that engage in little sand scattering may be immature, or even parasitising the efforts of others, and individual females may or may not be consistent from nesting to nesting. Also, an additional practical consideration is that measuring predation rates would encounter resistance from conservation organisations intent on reducing predation risks in the turtle nesting areas that they care for. We

agree it is an area worthy of investigation and a test of our ideas, though beyond the scope of our current work and practicalities. We have added text to the Discussion that acknowledges this referee's comment.

28) Pg 7 Line 45: Again, I'm not sure you can make this conclusions without at least mentioning that the sand scattering could be related to nest environment manipulation...

Agreed – please see above for note of this.

Appendix B

Manuscript ID RSOS-200327

Title - Buried treasure - Marine turtles do not 'disguise' or 'camouflage' their nests but avoid them and create a decoy trail

Authors - Thomas J. Burns, Rory R. Thomson, Rosemary A. McLaren, Jack Rawlinson, Euan McMillan, Hannah Davidson and Malcolm W. Kennedy

Accepted subject to minor changes – your email of 31st March 2020

Reviewer comments in regular script, our responses in italics.

Reviewer: 1

Comments to the Author(s)

General comments:

I thank the authors for taking the time to address my comments on their manuscript. I believe that the manuscript is clearer and more comprehensive following review and am happy that it is fit for publication following a few very minor changes (see below). For future reference, I encourage the authors to track their changes on the document or highlight regions that have been changed to facilitate a more effective way of gauging the changes that have been made in response to a reviewer's comments.

1) The inclusions of field sketches of the sand scattering stations are effective; however, they are excessive. I recommend keeping some 'type' examples that can be shown as examples and removing the rest.

We have reduced the total number of scattering path diagrams, retaining ten examples for each species that illustrate their diversity. The diagram for the two repeat return nesting hawksbill turtle is now only in the main paper.

Introduction

The introduction now reads much better, there are still a couple of missing references (detailed below), but I only have minor comments.

2) Lines 47-51: Insert a reference in regards to the nesting process, there should be one described in one of the volumes of The Biology of Sea Turtles if none others are available.

References added as suggested.

Methods

The methods are also clearer following the review, especially with the inclusion of the full statistical methodology in this section.

3) Lines 98-99: I appreciate the authors supplying the GPS coordinates for the study sites. I would recommend checking the GPS coordinates (especially 11.3166°N, -0.571031°E), keeping the number of decimal places consistent, and changing the longitudinal coordinates to 60.5600°W rather than having a negative Eastern coordinate.

Corrected the and formatted as suggested.

4) Lines 146-148: I think if you are going to use one of CCL or CCW, CCL makes more biological sense to include in the model. However, given that the analyses are completed and re-analyzing would cause a delay, I am content with CCW being used, but it's something to consider in the future.

Noted. We shall stick with CCW for this manuscript, and the two are highly correlated.

Results

The results are clear and concise and I have no further comments for improvement.

Discussion

4) Lines 275-278: I typically avoid and discourage the use of rhetorical questions, but leave it up to the discretion of the authors/editors as to whether it is rephrased into a statement.

We would like to keep this as a question.

5) Lines 289: Don't start a sentence with "And."

We have replaced 'And' with 'Furthermore', though there is no grammatical or literature objection to starting with 'And'.

Appendix C

Manuscript ID RSOS-200327

Accepted subject to minor changes

Title - Buried treasure - Marine turtles do not 'disguise' or 'camouflage' their nests but avoid them and create a decoy trail

Authors - Thomas J. Burns, Rory R. Thomson, Rosemary A. McLaren, Jack Rawlinson, Euan McMillan, Hannah Davidson and Malcolm W. Kennedy

Hello,

Many thanks for the good news about this paper. We have only been asked to make a few minor adjustments, all of which we have done – please see the response to comments file uploaded under "Response to Referees" in "Section 6 - File Upload" – file entitled 'Response to decision letter Burns et al 03Apr2020'. Please note that I have not included a contact telephone number for a press release because I will not be at my office during the coronavirus pandemic and do not wish my private number released, but I will be monitoring email regularly.

Best wishes,

Malcolm Kennedy

On behalf of all authors.

Malcolm W. Kennedy

Professor of Natural History

University of Glasgow

Glasgow G12 8QQ

Scotland

UK